# Genome-wide association study of resistance to *Mycobacterium tuberculosis* infection identifies a locus at 10q26.2 in three distinct populations

Jocelyn Quistrebert[1,2], Marianna Orlova[3,4,5], Gaspard Kerner[1,2,6], Le Thi Ton[7], Nguyễn Trong Luong[7], Nguyễn Thanh Danh[7], Quentin B. Vincent[1,2], Fabienne Jabot-Hanin[8,9,10], Yoann Seeleuthner[1,2], Jacinta Bustamante[1,2,11], Stéphanie Boisson-Dupuis[12], Nguyen Thu Huong[13], Nguyen Ngoc Ba[13], Jean-Laurent Casanova[1,2,12,14], Christophe Delacourt[2,15], Eileen G. Hoal[16], Alexandre Alcaïs[1,2], Vu Hong Thai[13], Lai The Thành[7], Laurent Abel[1,2,12‡], Erwin Schurr[3,4,5‡], Aurélie Cobat[1,2‡]*

1 Laboratory of Human Genetics of Infectious Diseases, Necker Branch, INSERM U1163, Paris, France, 2 University of Paris, Imagine Institute, Paris, France, 3 Infectious Diseases and Immunity in Global Health Program, Research Institute of the McGill University Health Centre, Montreal, Canada, 4 McGill International TB Centre, McGill University, Montreal, Canada, 5 Departments of Medicine and Human Genetics, McGill University, Montreal, Canada, 6 Human Evolutionary Genetics Unit, Institut Pasteur, UMR2000, CNRS, Paris, France, 7 Center for Social Disease Control, Binh Duong, Vietnam, 8 Bioinformatics Core Facility, Paris Descartes University, Paris, France, 9 Bioinformatics Platform, INSERM UMR U1163, Imagine Institute, Paris, France, 10 Necker Federative Research Structure, INSERM US24, CNRS UMS3633, Paris, France, 11 Study Center of Immunodeficiencies, Necker Hospital for Sick Children, Paris, France, 12 St Giles Laboratory of Human Genetics of Infectious Diseases, Rockefeller Branch, Rockefeller University, New York City, New York, United States of America, 13 Hospital for Dermato-Venereology, Ho Chi Minh City, Vietnam, 14 Howard Hughes Medical Institute, New YorkCity, New York, United States of America, 15 Paediatric Pulmonology and Allergology Department, Necker Hospital for Sick Children, Paris, France, 16 South African Medical Research Council Centre for Tuberculosis Research, DST-NRF Centre of Excellence for Biomedical Tuberculosis Research, Division of Molecular Biology and Human Genetics, Faculty of Medicine and Health Sciences, Stellenbosch University, Cape Town, South Africa

‡ These authors are joint senior authors on this work.
* aurelie.cobat@inserm.fr

**Data Availability Statement:** All summary statistics are available in the manuscript or at https://doi.org/10.5281/zenodo.3942125. Because

## Abstract

The natural history of tuberculosis (TB) is characterized by a large inter-individual outcome variability after exposure to *Mycobacterium tuberculosis*. Specifically, some highly exposed individuals remain resistant to *M. tuberculosis* infection, as inferred by tuberculin skin test (TST) or interferon-gamma release assays (IGRAs). We performed a genome-wide association study of resistance to *M. tuberculosis* infection in an endemic region of Southern Vietnam. We enrolled household contacts (HHC) of pulmonary TB cases and compared subjects who were negative for both TST and IGRA (n = 185) with infected individuals (n = 353) who were either positive for both TST and IGRA or had a diagnosis of TB. We found a genome-wide significant locus on chromosome 10q26.2 with a cluster of variants associated with strong protection against *M. tuberculosis* infection (OR = 0.42, 95%CI 0.35–0.49, $P$ = 3.71×10$^{-8}$, for the genotyped variant rs17155120). The locus was replicated in a French multi-ethnic HHC cohort and a familial admixed cohort from a hyper-endemic area of South Africa, with an overall OR for rs17155120 estimated at 0.50 (95%CI 0.45–0.55, $P$ = 1.26×10$^{-9}$). The variants are

of the IRB restriction on the data from Vietnam, France and South Africa, individual level data are only available in the context of a tuberculosis research project upon request from the VFSA TB infection Access Committee by contacting the committee chair Pr Stanislas Lyonnet (stanislas. lyonnet@inserm.fr), Director of the Imagine Institute in Paris, France. Stanislas Lyonnet was not involved with the research reported in this article.

**Funding:** This work was supported in part by the French Foundation for Medical Research (FRM) (EQU201903007798; FDM20170637664 to JQ), the Programme Hospitalier de Recherche Clinique (AOR-04-003); the Legs Poix (Chancellerie des Universités de Paris, https://www.sorbonne.fr/en/ to AA), the European Research Council (ERC-2010-AdG-268777 to LA), the National Institute of Allergy and Infectious Diseases of the National Institutes of Health (R01AI12434), the French National Research Agency (ANR) under the "Investments for the future" program (grant ANR-10-IAHU-01), the Integrative Biology of Emerging Infectious Diseases Laboratory of Excellence (ANR-10-LABX-62-IBEID), and the TBPATHGEN project (ANR-14-CE14-0007-01 to LA), the SCOR Corporate Foundation for Science (https://www.scor.com/fr/la-fondation-dentreprise-scor-pour-la-science to LA), the Rockefeller University (https://www.rockefeller.edu/), the Institut National de la Santé et de la Recherche Médicale (INSERM, https://www.inserm.fr/), Université de Paris (https://u-paris.fr/en/), the St. Giles Foundation (http://www.stgilesfoundation.org/), the Canadian Institutes of Health Research (CIHR, FDN-143332 to ES) and the Sequella/Aeras Global Tuberculosis Foundation (https://www.aeras.org, to ES). The funders had no role in study design, data collection and analysis, decision to publish, or preparation of the manuscript.

**Competing interests:** The authors have declared that no competing interests exist.

located in intronic regions and upstream of *C10orf90*, a tumor suppressor gene which encodes an ubiquitin ligase activating the transcription factor p53. *In silico* analysis showed that the protective alleles were associated with a decreased expression in monocytes of the nearby gene *ADAM12* which could lead to an enhanced response of Th17 lymphocytes. Our results reveal a novel locus controlling resistance to *M. tuberculosis* infection across different populations.

## Author summary

There is strong epidemiological evidence that a proportion of highly exposed individuals remain resistant to *M. tuberculosis* infection, as shown by a negative result for Tuberculin Skin Test (TST) or IFN-γ Release Assays (IGRAs). We performed a genome-wide association study between resistant and infected individuals, which were carefully selected employing a household contact design to maximize exposure by infectious index patients. We employed stringently defined concordant results for both TST and IGRA assays to avoid misclassifications. We discovered a locus at 10q26.2 associated with resistance to *M. tuberculosis* infection in a Vietnamese discovery cohort. This locus could be replicated in two independent cohorts from different epidemiological settings and of diverse ancestries enrolled in France and South Africa.

## Introduction

Tuberculosis (TB) remains a major public health threat worldwide [1]. An estimated 10 million people developed TB disease in 2018, of whom 1.45 million died. The causative agent of TB is *Mycobacterium tuberculosis* which is transmitted by aerosol from contagious TB patients. However, not all persons encountering infectious aerosols will become infected with *M. tuberculosis*, defining the first line of human resistance against TB [2,3]. Infection is inferred from the presence of anti-mycobacterial immunoreactivity, as shown by a positive result in tuberculin skin test (TST) and/or interferon-gamma (IFN-γ) release assay (IGRA). TST is done *in vivo* and consists of an intradermal injection of purified protein derivative (PPD) that provokes a delayed hypersensitivity reaction at the site of injection. IGRAs are performed *ex vivo* and measure the secretion of IFN-γ by leukocytes in response to *M. tuberculosis*-specific antigens. Both tests have their own limitations and results are not fully concordant [4–6]. Individuals who score positive by TST and/or IGRA are considered to suffer from asymptomatic latent TB infection (LTBI). Conversely, persons who score negative despite documented exposure to *M. tuberculosis* are considered resistant to infection. Based on TST and/or IGRA results, the intensity of exposure or the duration of follow-up, from 7% to 25% of subjects display the *M. tuberculosis* infection resistance phenotype [2,7–9].

The large inter-individual variability in exposure outcomes supports a major role for human genetic factors [10]. Various genome-wide approaches have confirmed this hypothesis by either considering TST and IGRA results as quantitative traits or relying on TST reactivity (positive/negative) as a surrogate marker for infection[11–14]. Regarding this latter phenotype, persistent TST negativity was linked to loci on 2q21-2q24, further fine-mapped to *ZEB2*, and 5p13-5q22 in an Ugandan population [14,15]. A major locus, named *TST1* on chromosome 11p14, employing stringently defined TST negativity (0 mm vs. > 0 mm) as phenotype, was identified in a linkage analysis conducted in South Africa [11]. *TST1* was later replicated in a

household contact (HHC) study of French families [16]. A genome-wide association study (GWAS) among highly *M. tuberculosis* exposed HIV-seropositive individuals from East Africa, identified a locus in the 5q31.1 region near *IL9* associated with negative TST [17].

Here, we performed a GWAS of resistance to *M. tuberculosis* infection using a robust phenotype based on both TST and IGRA information. In addition, we used a HHC study design guaranteeing shared environmental effects and high intensity of exposure to *M. tuberculosis*. We found a locus on chromosome 10q26.2 associated with resistance to *M. tuberculosis* infection in an East Asian population from Southern Vietnam. Importantly, this locus was replicated in two other cohorts from France and South Africa, representing different ancestries and epidemiological settings.

## Results

### Genome-wide association study of resistance to *M. tuberculosis* infection in Vietnam

First, we conducted a GWAS in the Vietnamese sample using 185 uninfected and 353 infected subjects, consisting of 201 individuals with positive TST/QuantiFERON-TB Gold In-Tube test (QFT-GIT) results and 152 pulmonary TB (PTB) patients. A total of 5,591,951 high quality variants were tested with a genomic inflation factor ($\lambda$) at 0.997, suggesting that effects from the familial study design were well controlled (S1 Fig). The corresponding Manhattan plot is shown in Fig 1A. We observed a genome-wide significant association on chromosome 10q26.2, corresponding to a cluster of 12 variants and 6 additional variants in high linkage disequilibrium (LD) with $P < 5 \times 10^{-7}$ in the intronic regions and upstream of *C10orf90* (or *FATS*, HGNC: 26563) (Fig 1B). The top-associated variant was the imputed rs11245088 (odds ratio (OR) = 0.42, 95% confidence interval (CI) 0.39–0.45, $P = 1.58 \times 10^{-8}$) while the top-associated genotyped variant was rs17155120 ($P = 3.71 \times 10^{-8}$) (Table 1). Each copy of the minor allele T of rs17155120 conferred protection against *M. tuberculosis* infection with an OR of being infected for CT vs. CC or TT vs. CT at 0.42 (95%CI 0.35–0.49) (Fig 1C). The intensity cluster plot for rs17155120 showed that the genotype calling was of high quality and separated clearly into 3 genotype groups (S2 Fig). Since all 18 variants in the locus were in high LD (S3 Fig), the imputed variants were likely to have a high imputation quality as suggested by their info score (S1 Table).

We also performed a GWAS between the 185 uninfected and the 201 infected subjects, excluding the 152 PTB patients (S4 Fig). Despite a smaller sample size, all the 18 variants of the locus were still associated with protection against infection with $P < 5.0 \times 10^{-6}$, with similar ORs (for rs17155120, OR = 0.40, 95%CI 0.32–0.49, $P = 2.55 \times 10^{-7}$) (S2 Table). Similar findings were also observed when considering only the 152 PTB patients as infection reference, with an OR for rs17155120 estimated at 0.50 (95%CI 0.41–0.59, $P = 2.10 \times 10^{-4}$). These results indicate that PTB patients are an appropriate infection reference group in this analysis.

### Replication of variants associated with resistance to *M. tuberculosis* infection in France and South Africa

We tested the effects of the variants of the 10q26.2 locus in a French multi-ethnic HHC cohort (30 uninfected vs. 157 infected subjects) and an admixed familial sample from South Africa (118 uninfected vs. 136 infected subjects). In the French cohort, 17 variants of the cluster could be genotyped or imputed (including 11 genome-wide significant variants), and 3 were replicated at the $P < 0.025$ level with effect sizes in the same direction as in the Vietnamese cohort. The most significant associated variant was rs56106518 (OR = 0.40, 95%CI 0.30–0.51,

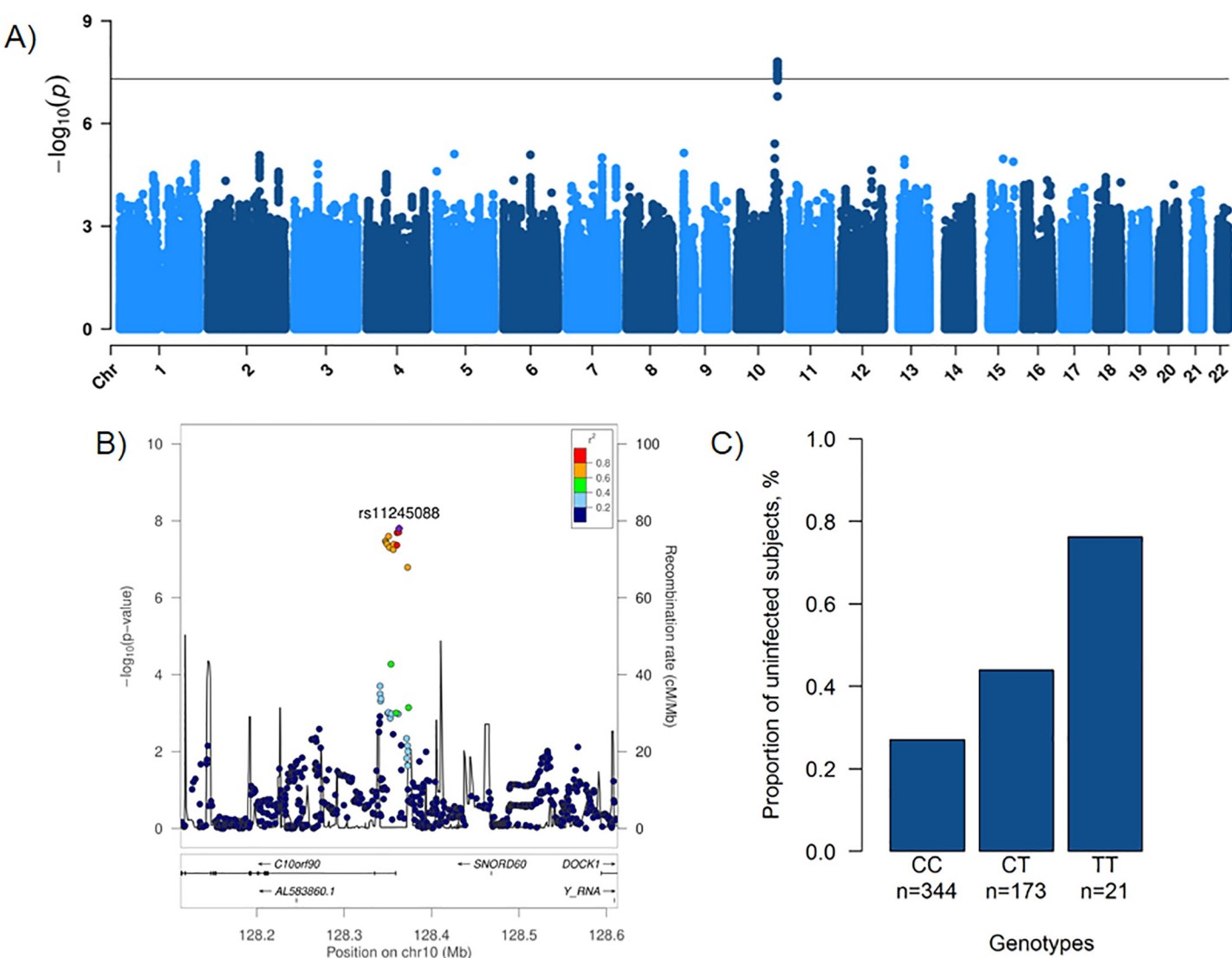

**Fig 1. Genome-wide association study of resistance to *M. tuberculosis* infection in Vietnam. A)** Manhattan plot showing results from a genome-wide association study between 185 uninfected subjects (negative for both tuberculin skin test and QuantiFERON-TB Gold In-Tube test) and 353 infected subjects (201 infected individuals positive for both tests and 152 patients with a history of pulmonary tuberculosis) for 5,591,951 variants (minor allele frequency > 5% and info > 0.8) with an unadjusted additive genetic model. The $-\log_{10}(P$ value) for each variant (y-axis) is presented according to its chromosomal position (x-axis, build hg19). The dashed line indicates the genome-wide significant threshold at $P = 5 \times 10^{-8}$. **B)** Locus zoom plot showing association for the 10q26.2 locus, in a 500 kb window surrounding the top imputed variant rs11245088 (purple diamond). Colors represent pairwise linkage disequilibrium ($r^2$) with rs11245088 as calculated for the Vietnamese Kinh population of 1000 Genomes phase 3. **C)** Proportion of Vietnamese individuals resistant to *M. tuberculosis* infection by genotype for the variant rs17155120. Each bar represents the proportion of uninfected subjects among CC individuals (n = 93/344), CT individuals (n = 76/173) and TT individuals (n = 16/21) for the variant rs17155120 in Vietnam.

$P = 2.98 \times 10^{-3}$) (Table 1). In the South African population, 12 variants of the locus were geno-typed or successfully imputed (including 10 genome-wide significant variants) and 3 of them showed evidence for replication. The most significant associated variant was rs118037357 (OR = 0.55, 95%CI 0.43–0.67, $P = 7.38 \times 10^{-3}$) (Table 1). The top genotyped variant rs17155120 was also replicated in the two cohorts ($P_{France} = 1.51 \times 10^{-2}$ and $P_{SouthAfrica} = 1.74 \times 10^{-2}$) (Table 1 and Fig 2A).

Interestingly, the variants across the two replication cohorts were in high LD with each other presenting similar LD patterns as in Vietnam (S5 Fig). The frequencies of the

**Table 1. Association between an additive genetic effect of variants on chromosome region 10q26.2 and resistance to *M. tuberculosis* infection in Vietnam, France and South Africa.**

| Variant | *C10orf90* | EA | Vietnam | | | France | | | South Africa | | |
|---|---|---|---|---|---|---|---|---|---|---|---|
| | | | EAF | OR (95%CI) | *P* value | EAF | OR (95%CI) | *P* value | EAF | OR (95%CI) | *P* value |
| rs11245088 | upstream | C | 0.25 | 0.42 (0.39–0.45) | $1.58\times10^{-8}$ | - | - | - | 0.44 | 0.99 (0.91–1.07) | $4.75\times10^{-1}$ |
| rs72163291 | intron | ins | 0.24 | 0.42 (0.35–0.49) | $1.94\times10^{-8}$ | 0.37 | 0.88 (0.78–0.98) | $3.54\times10^{-1}$ | - | - | - |
| rs17155143 | upstream | A | 0.20 | 0.39 (0.31–0.47) | $1.98\times10^{-8}$ | 0.18 | 0.63 (0.50–0.77) | $1.29\times10^{-1}$ | 0.15 | 0.60 (0.48–0.72) | $1.72\times10^{-2}$ |
| rs7909756 | upstream | G | 0.24 | 0.42 (0.35–0.49) | $2.03\times10^{-8}$ | 0.37 | 0.83 (0.73–0.93) | $2.87\times10^{-1}$ | 0.36 | 0.90 (0.81–0.99) | $2.77\times10^{-1}$ |
| rs28703703 | intron | G | 0.20 | 0.41 (0.34–0.49) | $2.52\times10^{-8}$ | 0.20 | 0.59 (0.48–0.69) | $4.95\times10^{-2}$ | 0.17 | 0.66 (0.55–0.77) | $2.95\times10^{-2}$ |
| rs56106518 | intron | C | 0.23 | 0.41 (0.33–0.48) | $3.07\times10^{-8}$ | 0.30 | 0.40 (0.30–0.51) | $2.98\times10^{-3}$ | - | - | - |
| rs75482972 | intron | A | 0.20 | 0.42 (0.34–0.49) | $3.35\times10^{-8}$ | 0.20 | 0.58 (0.47–0.68) | $4.38\times10^{-2}$ | 0.17 | 0.68 (0.57–0.78) | $3.48\times10^{-2}$ |
| rs17155120 | intron | T | 0.20 | 0.42 (0.35–0.49) | $3.71\times10^{-8}$ | 0.18 | 0.48 (0.36–0.59) | $1.51\times10^{-2}$ | 0.16 | 0.62 (0.52–0.73) | $1.74\times10^{-2}$ |
| rs73370887 | intron | A | 0.20 | 0.40 (0.33–0.48) | $4.05\times10^{-8}$ | 0.31 | 0.78 (0.71–0.85) | $1.43\times10^{-1}$ | 0.28 | 0.75 (0.67–0.83) | $3.69\times10^{-2}$ |
| rs79608098 | intron | T | 0.20 | 0.42 (0.35–0.49) | $4.06\times10^{-8}$ | 0.20 | 0.58 (0.48–0.69) | $4.50\times10^{-2}$ | 0.17 | 0.66 (0.55–0.76) | $2.71\times10^{-2}$ |
| rs61750007 | upstream | C | 0.24 | 0.43 (0.36–0.50) | $4.30\times10^{-8}$ | 0.24 | 0.53 (0.42–0.64) | $3.49\times10^{-2}$ | 0.20 | 0.73 (0.62–0.84) | $7.62\times10^{-2}$ |
| rs77513326 | intron | A | 0.20 | 0.42 (0.34–0.49) | $4.93\times10^{-8}$ | 0.17 | 0.47 (0.36–0.59) | $1.61\times10^{-2}$ | 0.16 | 0.63 (0.52–0.75) | $2.35\times10^{-2}$ |
| rs79918233 | intron | A | 0.20 | 0.41 (0.33–0.49) | $5.61\times10^{-8}$ | 0.17 | 0.51 (0.39–0.63) | $3.00\times10^{-2}$ | 0.16 | 0.63 (0.52–0.74) | $1.95\times10^{-2}$ |
| rs147584264 | upstream | C | 0.19 | 0.41 (0.33–0.49) | $1.21\times10^{-7}$ | 0.22 | 0.77 (0.62–0.93) | $3.00\times10^{-1}$ | - | - | - |
| rs191820708 | upstream | A | 0.19 | 0.41 (0.33–0.49) | $1.31\times10^{-7}$ | 0.19 | 0.63 (0.50–0.77) | $1.40\times10^{-1}$ | - | - | - |
| rs201178890 | upstream | T | 0.19 | 0.41 (0.33–0.49) | $1.36\times10^{-7}$ | 0.22 | 0.79 (0.62–0.97) | $3.36\times10^{-1}$ | - | - | - |
| rs202189321 | upstream | T | 0.20 | 0.41 (0.33–0.49) | $1.37\times10^{-7}$ | 0.18 | 0.65 (0.52–0.78) | $1.44\times10^{-1}$ | - | - | - |
| rs118037357 | upstream | A | 0.19 | 0.41 (0.33–0.49) | $1.62\times10^{-7}$ | 0.15 | 0.47 (0.33–0.60) | $2.83\times10^{-2}$ | 0.14 | 0.55 (0.43–0.67) | $7.38\times10^{-3}$ |

CI, confidence intervals; EA, effect allele; EAF, effect allele frequency; OR, odds ratio; ins, insertion

rs17155120 T allele for all 3 cohorts were also similar, ranging from 0.16 to 0.20. The rs17155120 T allele frequency of 0.20 in the Vietnamese cohort was consistent with the Kinh allele frequency of 0.18 in 1000 Genomes (1000G) phase 3 (S6 Fig [18], and S3 Table). The rs17155120 T allele frequency of 0.18 in the multi-ethnic French cohort was also close to the global frequency of 0.16 in all 1000G populations. However, the frequency of 0.16 for rs17155120 T allele in the Souh African cohort was higher than in any 1000G African population that ranged from 0.05 to 0.11. Discrepancy of allele frequencies between Souh African subjects and African populations of 1000G was confirmed by different patterns of LD in a 30 kb region around rs17155120 (S7 Fig), which could be explained by the specific ethnic origin of the Souh African subjects.

## Trans-ethnic meta-analysis

Next, we performed a meta-analysis of resistance to *M. tuberculosis* infection using the GWAS data of the 3 cohorts. The combined analysis was carried out in 333 uninfected and 616 infected subjects. The only genome-wide significant result was observed with the variants of the 10q26.2 region (Figs 2B and S8). The most significant signal was observed at the genotyped variant rs17155120 (OR = 0.50, 95%CI = 0.45–0.55, $P = 1.26 \times 10^{-9}$) and no heterogeneity was observed across the 3 studies ($P_{het} = 0.31$) (Fig 2A and S4 Table).

## Functional annotation

The variants of the 10q26.2 region map to the intronic regions and upstream of the tumor suppressor gene *C10orf90* (or *FATS*, for Fragile-site Associated Tumor Suppressor, HGNC: 26563) (Fig 3). According to ENCODE, two associated intronic variants, rs28703703 and

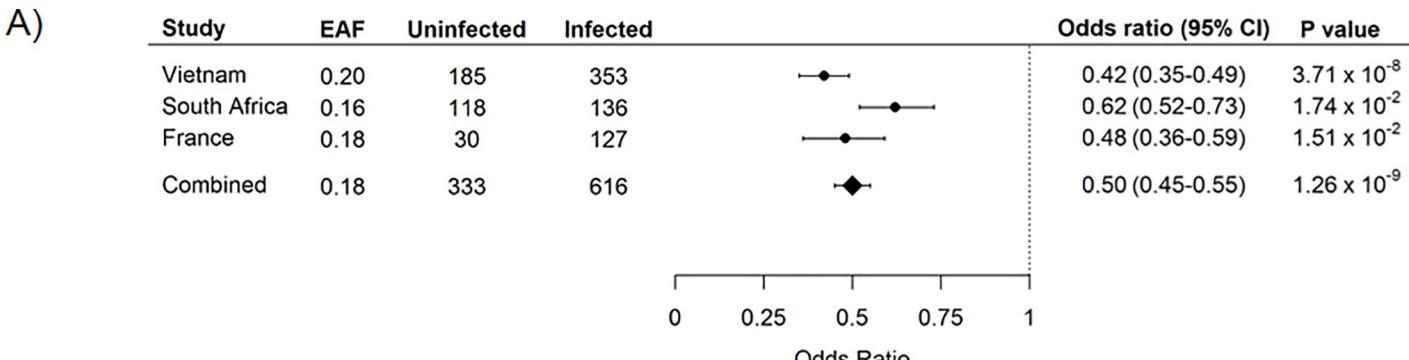

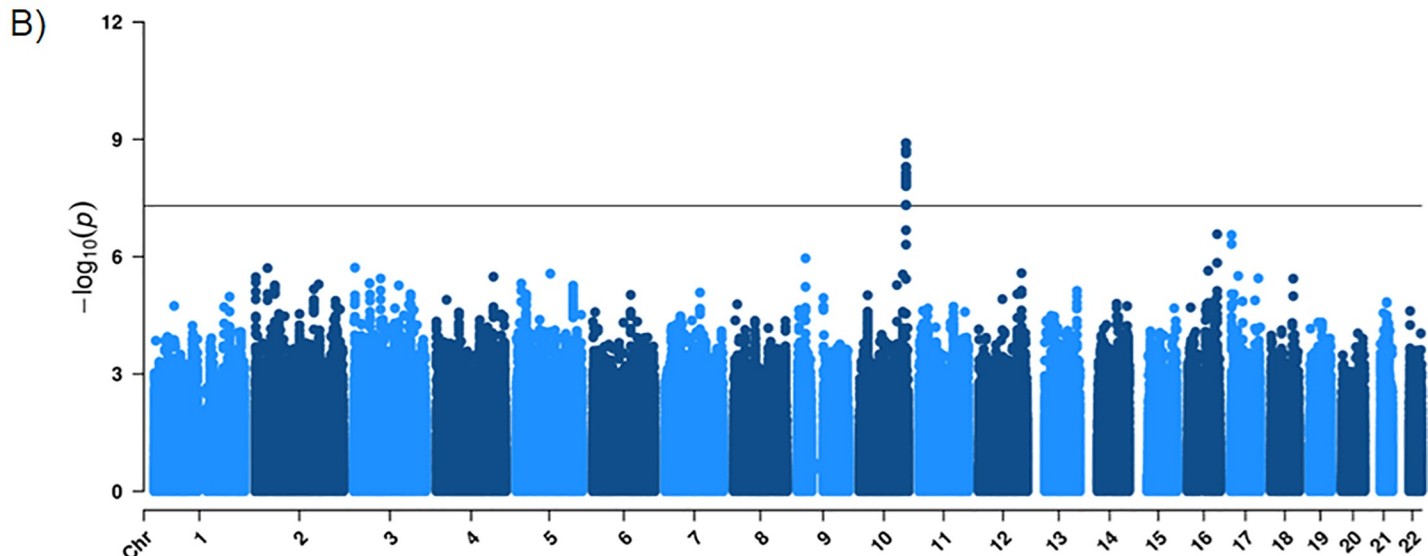

**Fig 2. Genome-wide association study of resistance to *M. tuberculosis* infection in 3 cohorts from Vietnam, France and South Africa. A)** Forest plot of the association between an additive genetic effect of rs17155120 on chromosome region 10q26.2 and resistance to *M. tuberculosis* infection. Odds ratios and 95% confidence intervals derived from a linear mixed model, *P* values, sample sizes and frequency of the effect allele (EAF) are reported by individual cohort and for the random effects meta-analysis. **B)** Manhattan plot showing results from a genome-wide association study between 333 uninfected subjects and 616 infected subjects for 3,967,482 variants (minor allele frequency > 5% and info > 0.8) with an unadjusted additive genetic model. The $-\log_{10}(P$ value) for each variant (y-axis) is presented according to its chromosomal position (x-axis, build hg19). The dashed line indicates the genome-wide significant threshold at $P = 5 \times 10^{-8}$.

rs77513326, are located in a regulatory genomic region characterized by H3K4me1 and H3K27ac histone marks and an active enhancer signature in lymphocytes T helper 17 (Th17). The variant rs77513326 also overlaps ATAC peaks in Th17 cells, memory T cells, natural killer cells and CD8[+] T cells. We further explored the variants in various expression quantitative trait loci (eQTLs) databases of relevant tissues for the phenotype and found an association between them and expression of the nearby gene *ADAM12*. In particular, rs28703703 and the genotyped variant rs17155120 displayed decreasing expression of *ADAM12* with each minor allele (having a protective effect against *M. tuberculosis* infection) in monocytes ($P = 2.10 \times 10^{-3}$ and $P = 4.70 \times 10^{-3}$ respectively) (from https://immunpop.com/kim/eQTL [19]). No association was observed in other immune cell types.

## Discussion

In this study, we explored the genetic determinants of natural resistance to *M. tuberculosis* infection after intense exposure. There are no direct tests for established infection because TST

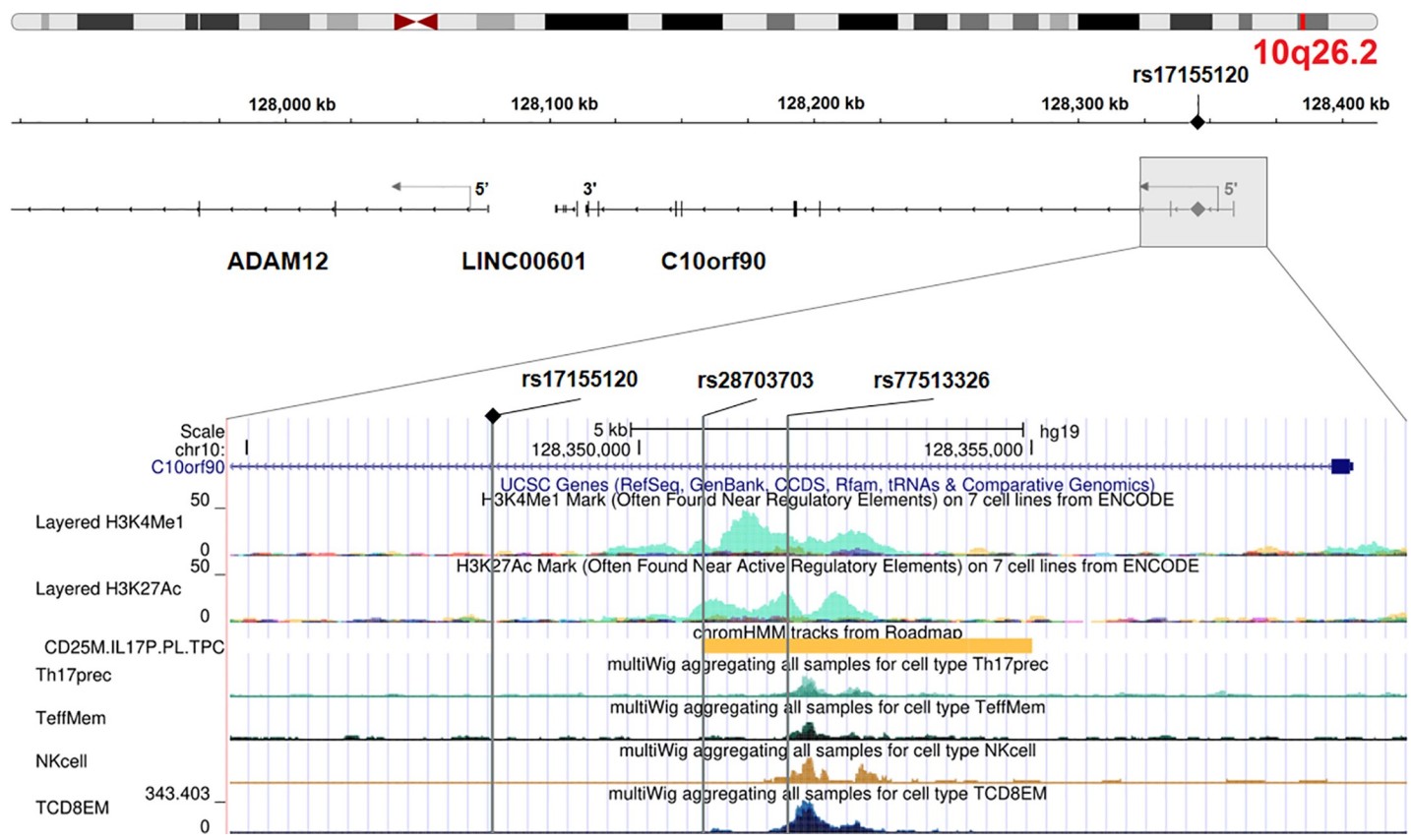

**Fig 3. Genomic annotation of the locus on chromosome region 10q26.2.** The upper panel is adapted from Integrative Genomics Viewer (http://igv.org/app/) and the lower panel is adapted from UCSC Genome Browser (http://genome.ucsc.edu/). The 3 vertical grey lines represent the associated variants rs1715120, rs28703703 and rs77513326 (from left to right) that overlap regulatory regions. From top to bottom: H3K4me1 and H3K27ac histone marks from ENCODE, active enhancer in Th17 cells (chromHMM annotation from ROADMAP), chromatin accessibility as represented by ATAC peaks in Th17 cells, memory T cells, NK cells and CD8+ T cells [20].

and IGRAs measure an immune response that does not allow to distinguish between past or present infection with *M. tuberculosis* bacilli. Nevertheless, our results show a strong genetic effect on resistance to *M. tuberculosis* infection irrespective of infection being persistent or temporary. TST and IGRA tests have well-known limits [4]. To minimize misclassification of uninfected and infected subjects in a cross-sectional setting, we therefore relied on negative results in both tests and defined stringent cut-offs. As exposure to *M. tuberculosis* is difficult to quantitate, yet a critical feature of TB studies [21], we focused on individuals at high risk of infection. We recruited household contacts recently exposed for extended periods to a contagious PTB index, some of whom remained infection-free. We discovered a genome-wide significant association between a cluster of variants on chromosome 10q26.2 and resistance to *M. tuberculosis* infection in well-characterized Vietnamese subjects. Strikingly, this locus could be replicated in two independent cohorts with different epidemiological settings from France and South Africa, resulting in association of the C/T variant rs17155120 across the 3 populations at an estimated combined odds ratio of 0.50 (95%CI 0.45–0.55) for becoming infected for TC vs. CC or TT vs. TC individuals.

The cluster of associated variants overlaps intronic and 5' regions of *C10orf90* (or *FATS*, HGNC: 26563), a tumor suppressor gene. This gene lies within a common-fragile site, which is an evolutionarily conserved region among mammals and susceptible to DNA damage [22]. The protein encoded by *C10orf90* has been shown to promote p53 activation in response to

DNA damage through an E3 ubiquitin ligase activity [23,24]. Several E3 ubiquitin ligases have already been shown to participate in the defense against *M. tuberculosis*, in particular through autophagy[25–27]. The p53 transcription factor is not only a master regulator of autophagy but also activates apoptosis, another key process for host infected cells to limit the spread of pathogens as *M. tuberculosis* [28,29]. Recent studies demonstrated that p53-induced apoptosis plays a critical role in the inhibition of mycobacteria survival and the macrophage resistance, possibly mediated by IL-17 [30,31]. However, possible role of *C10orf90* during *M. tuberculosis* infection or any related immune process remains unknown.

*In silico* functional annotation of the 10q26.2 associated variants revealed additional regulatory features related to lymphocytes Th17. In particular, the A/G variant rs28703703 was identified as a likely cis-eQTL for *ADAM12* in monocytes, with decreased expression of *ADAM12* associated with the G allele, protective against *M. tuberculosis* infection. *ADAM12* is located ~280 kb downstream *C10orf90*, and encodes a matrix metalloprotease linked to a broad range of biological processes [32]. *ADAM12* expression correlates with lung inflammation as it is overexpressed in cells issued from asthmatic sputum and in the airway epithelium during allergic inflammatory reaction [33,34]. Previous studies also reported *ADAM12* expression in Th17 cells [35,36], and *ADAM12* knockdown in human T cells was found to increase Th17 cytokine production (IL-17A, IL-17F, and IL-22) [35]. In South India, TST-negative individuals produced significantly higher levels of Th17 cytokines than TST-positive individuals [37]. Similarly, significant higher levels of Th17 cytokines were observed in persistent negative IGRA individuals as compared with IGRA converters in a recent study conducted in the Gambia [38]. These findings are consistent with our present results showing that the rs28703703 G allele protective against *M. tuberculosis* infection is associated with lower *ADAM12* expression that could lead to higher Th17 cytokine production. Overall, these observations support the view that Th17 cytokines may have a protective role against early stages of *M. tuberculosis* infection.

The 3 enrolled samples were of modest size. However, the design of the 3 studies, and the definition of a stringent and homogenous phenotype across them enabled us to detect a significant association with large effects. Interestingly, adding the PTB Vietnamese patients, who are by definition infected, increased the power of the analysis. Strikingly, the top associated variants displayed large effects with similar allele frequencies across populations and were in high LD across multiple ancestries. This observation was not expected because of the various epidemiological settings and the genetic diversity of the cohorts that included Kinh Vietnamese, Europeans, North Africans, Sub-Saharan Africans and admixed Western Cape Coloureds. Such populations are usually under-represented in studies of genetic association while they could provide valuable insights to understand complex diseases [39].

In conclusion, we demonstrated that rigorous epidemiological design and phenotype definition with seemingly limited sample sizes can reveal novel genetic factors that offer protection from major pathogens such as *M. tuberculosis*. We found that *C10orf90* and *ADAM12* are promising candidate genes involved in the natural resistance to *M. tuberculosis*. Further investigations are needed to elucidate their role in the process of the initial infection, which could be a major step to provide new opportunities in the fight against TB.

## Subjects and methods

### Ethics statement

Signed informed consent was obtained from all the participants, and from the parents of enrolled minors. The study was approved by the regulatory authorities in Binh Duong, Vietnam (1366/UBND-VX), the French Consultative Committee for Protecting Persons in

Biomedical Research of Henri Mondor Hospital (Créteil, France) and the Stellenbosch University Health Research Ethics Committee (Tygerberg, South Africa).

## Study settings and populations

**Vietnam.** Vietnam is a middle-income country in South-East Asia with a high annual incidence of TB (130/100,000 at the time of the study [40]), high Bacille Calmette-Guérin (BCG) vaccination coverage (with reported rates exceeding 95% [41]) and a low population prevalence of HIV (0.4% in 2015 in the general population and 5% among TB cases [40,42]). From 2010 to 2015, we recruited, in a *M. tuberculosis* endemic region of Southern Vietnam, index PTB adults with persistent cough before the start of treatment (mean duration of coughing = 3.2 months). PTB diagnosis was assessed by clinical presentation, chest X-ray and/or positive cultures for *M. tuberculosis*. A total of 1108 HHCs of 466 PTB index cases were invited to participate in the study and underwent both TST and QFT-GIT (S1 Text). HHCs did not undergo any further follow-up.

**France.** As a first replication cohort, we used household TB contacts studied in Val-de-Marne, a suburban region of Paris, in the context of a general screening procedure. This multi-ethnic cohort has been previously described [12,16]. Val-de-Marne is an area of low TB endemicity, displaying an annual incidence of 22/100,000 at the time of the study [43] and BCG vaccination rates are high [41]. HIV seroprevalence is low in France in the general population (0.3% at the time of the study [44]) and was estimated at 7% in the index cases of the study [45]. For this study, 664 HHCs of 132 PTB index cases were investigated according to national guidelines that required two screening visits. HHCs were individuals sharing residence with an index during the 3 months before diagnosis. Briefly, the first visit (V1) included a physical examination, a chest radiograph, TST and in-house IGRA [12,16] (S1 Text). These investigations, except for IGRA, were repeated 8–12 weeks later (V2) if the contact subject did not meet the criteria for infection at V1.

**South Africa.** As a second replication cohort, we used a large sample (n = 415) of 153 nuclear families from a suburban area of Cape Town, South Africa, which has been previously described [6,11,12,16]. All individuals belonged to the South African Coloured group, a unique multi-way admixed population [46]. There was no specific requirement for subjects to be HHCs of PTB patients. Indeed, TB is hyper-endemic in this area with an incidence of ~800/100,000 at the time of the study [47] and TB transmission occurs more often outside the household [48]. BCG vaccination at birth is routine in this area [49]. HIV seroprevalence was estimated at 5.2% in the overall population and less than 2% in the pediatric population at the time of the study [11]. In addition, individuals who were known to be HIV positive, pregnant, or using immunomodulatory chemotherapy were excluded at the time of enrollment [11]. TST and in-house IGRA were performed as previously described (S1 Text). Subjects who had clinical TB disease in the two years preceding the study were excluded. Thus, only healthy children and young adults from the area were included and tested for infection.

## Genotyping, quality control and imputation

A total of 724 individuals from Vietnam (S9 Fig) and 573 from France were genotyped using the Illumina Infinium OmniExpressExome-8-v1 chip (960,212 single nucleotide polymorphisms, SNPs). For the South African cohort (n = 374), the Illumina HumanOmni2.5–8 Bead-Chip (~2 million of SNPs) was used. All quality control steps were done in each cohort with PLINK v1.9 [50]. Autosomal SNPs with a minor allele frequency (MAF) > 0.01, a genotype call rate > 0.99 and a Hardy-Weinberg (HWE) equilibrium $P > 1.00 \times 10^{-5}$ were retained. Individuals with a call rate < 95% were excluded (n = 2). Identity-by-descent (IBD) analysis

was done to detect duplicated individuals and the members of the pairs with the lower call rate were excluded (n = 1). After the quality control, imputation was performed on 720 individuals and 598,090 variants from Vietnam and 573 individuals and 886,471 variants from France using the Michigan Imputation Server [51] with Eagle2 [52] for the pre-phasing and the 1000G Project as reference panel [53]. For the South African cohort (374 individuals and 1,347,846 variants), the imputation was done on the Sanger Imputation Server with Eagle2 for the pre-phasing and the African Genome Resources as reference panel which includes ~2000 African samples in addition to the individuals from 1000G [54]. Imputed SNPs with an imputation quality info score > 0.8 and MAF > 0.05 were retained for further association analyses (5,591,951 variants in Vietnam, 7,737,070 variants in France and 6,922,541 variants in South Africa).

For each cohort, principal component analysis (PCA) was conducted to evaluate population structure. Genotypes of the individuals from 1000G were used to calculate principal components and data for subjects from the cohorts were projected onto the eigenvectors. The Vietnamese cohort, which was sampled from the Vietnamese Kinh group, was homogenous and clustered with the 1000G East Asian populations (S10 Fig). By contrast, the families in the Val-de-Marne sample showed genetic diversity at the population level with a majority of individuals of European, North African and Sub-Saharan ancestries (S11 Fig). The admixed Souh African subjects, who exhibited genetic diversity at the individual level, were forming a distinct cluster close to the African populations of 1000G (S12 Fig).

## Definition of the *M. tuberculosis* infection phenotype

The definition of the *M. tuberculosis* infection phenotype relied on both TST and IGRA results. In particular, we used a 5 mm cut-off to determine TST status, based on previous studies in similar settings [7,17] and published guidelines [55]. We explored covariates associated with our infection definition in the entire cohort of enrolled individuals and the subset of those with genotype information were retained for the GWAS (S9 Fig).

For the Vietnamese study, resistance to *M. tuberculosis* infection was defined by the presence of a negative TST < 5mm and a negative QFT-GIT test result following a protocol suggested by the manufacturer (S1 Text). Infected individuals were defined as subjects presenting both a positive TST ≥ 5mm and a positive QFT-GIT test result. A total of 188 subjects were classified as double negative and 512 as double positive (S13 Fig), among which 185 and 201 subjects were genotyped, respectively. In order to increase the sample size, we also added 152 genotyped PTB patients, consisting of 146 index cases and 6 subjects with a history of PTB, to the infected group (S5 Table). We investigated covariates associated with our infection definition and no significant association was found (S6 Table). Therefore, we conducted an unadjusted genetic association analysis of our binary infection phenotype.

For the French study, which included only HHCs and none of their PTB index cases, contacts could have had one or two screening visits (V1 and V2) with a TST measurement (S14 Fig). A TST was considered negative when the skin induration was (i) < 5 mm at both V1 and V2, (ii) < 5 mm at V1, when only one visit was done. A TST was considered positive when the skin induration was (i) ≥ 5 mm at both V1 and V2, (ii) < 5 mm at V1 and ≥ 10 mm at V2, which reflected true conversions. In-house IGRA was used in this study and provided quantitative levels of IFN-γ production upon early secretory antigenic target 6 (ESAT6) stimulation (S1 Text). A negative IGRA result was defined by a null production of IFN-γ. To determine the optimal positivity cut-off, we built a receiver operating characteristic (ROC) curve with TST status as the observed outcome and the corrected IFN-γ levels (ESAT6 response minus non-stimulated control value) as the predicted outcome among all the contacts enrolled. We

selected as positivity threshold the highest sum of sensitivity plus specificity, which was equal to 175 pg/mL (S14 Fig). Then, we defined uninfected subjects as HHCs with a negative TST and a null IFN-γ production (n = 33) and infected subjects as HHCs who presented both a positive TST and a positive IGRA result (IFN-γ production > 175 pg/mL) (n = 147) (S14 Fig). We also looked for covariates associated with our infection definition in this sample of 180 individuals (S7 Table). Age was the only factor significantly associated with our infection definition and was included as covariate in the genetic association analysis that finally included 30 genotyped uninfected and 127 infected subjects (S8 Table).

For the South African study, a 5 mm cut-off was used to identify negative and positive TSTs (S15 Fig). In-house IGRA was also based on the production of IFN-γ upon ESAT6 stimulation (S1 Text). We determined the optimal IGRA positivity cut-off similarly to the French study by building a ROC curve, leading to a threshold of 20.9 pg/mL (S15 Fig). Then, we defined uninfected subjects as HHCs with a negative TST (< 5 mm) and a null IFN-γ production (n = 128), and infected subjects as those with both positive TST and IGRA result (IFN-γ production > 20.9 pg/mL) (n = 152) (S15 Fig). Age was the only factor significantly associated with our infection definition in the whole cohort (S9 Table) and was included as covariate in the genetic association analysis that finally included 118 genotyped uninfected and 136 infected subjects (S10 Table).

Covariates associated with our infection definition were investigated using mixed-effects logistic regression with a random effect per family in each cohort. All the analyses were carried out using R software (version 3.5.2) and related packages "pROC" and "lme4"[56–58].

### Genetic association analyses

We conducted genetic association analyses of uninfected vs. infected subjects in the 3 cohorts using a linear mixed-model (LMM) assuming an additive genetic model as implemented in GEMMA v0.98 [59]. To account for the familial relationships, a genetic relatedness matrix (GRM) was used as random effects. In each cohort, the GRM was estimated using centered genotypes after the quality control described above. *P* values from the likelihood ratio test were reported. For better interpretability, we reported odds ratios (OR) and their 95% confidence intervals (95%CI) after transforming the regression coefficients of the LMM [60]. Manhattan plots of the -$\log_{10}(P$ values) and quantile-quantile (QQ) plots were generated using "CMplot" package in R [61]. Regional plots were generated using LocusZoom Standalone v1.4 [62]. Haplotype plots were generated using Haploview [63]. Replication of genome-wide associated variants ($P < 5 \times 10^{-8}$) in the primary cohort from Vietnam was assessed in the two cohorts from France and South Africa. The observed LD in France and South Africa was slightly weaker as compared with Vietnam, and two LD blocks were inferred by Haploview (S5 Fig). We therefore considered variants at a nominal one sided *P* value < 0.025 and with a consistent direction of the effect size as replicated.

We also conducted a trans-ethnic meta-analysis by using summary statistics (i.e. beta estimates and their standard errors) from the Vietnamese discovery cohort and the two replication datasets. We used the random-effects model of Han and Eskin implemented in METASOFT [64]. This model assumes effect sizes of exactly zero in all the studies (i.e. no heterogeneity) under the null hypothesis of no associations and allows the effect sizes to vary among studies (i.e. heterogeneity) under the alternative hypothesis. The effect size consistency across studies were determined using the Cochran's Q statistic. Allelic effect estimates were also derived on the log-odds scale.

### Functional annotation

We used the UCSC Genome Browser [65] to identify associated variants which may overlap with known regulatory regions: 1) histone marks from the ENCODE project [66], 2)

chromatin state annotated by ChromHMM on the basis of ROADMAP [67,68] and 3) chromatin accessibility determined by assay for transposase-accessible chromatin using sequencing (ATAC-seq) from immune cell-types [20]. We also looked at the associated variants in eQTL databases which focused on gene expression in monocytes [19,69], T cells [70], macrophages [71], and various types of immune cells [72].

## Supporting information

**S1 Text. Supplemental Methods.** Immune assays in Vietnam, France and South Africa.
(PDF)

**S1 Fig. Quantile-quantile plot of GWAS resistance to *M. tuberculosis* infection in Vietnam.**
Plot of the expected distribution of association test statistics (x axis) for 5,591,951 variants compared to the observed values (y axis). Red line is the null hypothesis of no association (y = x).
(PDF)

**S2 Fig. Intensity cluster plot for rs17155120 of the Illumina Infinium OmniExpressExome-8-v1 chip in the 720 Vietnamese individuals who were genotyped.**
(PDF)

**S3 Fig. Haploview LD graph of the top associations on chromosome 10q26.2 in Vietnam.**
Pairwise LD coefficients $r^2$ are shown in each cell between 18 variants with $P < 5.0 \times 10{-}6$. Top genotyped variant rs17155120 is second from left.
(PDF)

**S4 Fig. Manhattan plot for the GWAS of resistance to *M. tuberculosis* infection in Vietnam (185 uninfected vs 201 infected participants).**
(PDF)

**S5 Fig. Haploview LD graphs of the locus on chromosome 10q26.2 in replication cohorts.**
Pairwise LD coefficients $r^2$ are shown in each cell A) between the 17 variants of the locus in the French cohort, and B) between 12 out of the 18 variants of the locus in the South African cohort. Top genotyped variant rs17155120 is second from left in each figure.
(PDF)

**S6 Fig. Geographic distribution of the variant rs17155120 in 1000G phase 3 populations.**
The T allele (in blue) is protective against *M. tuberculosis* infection. Map generated from the Geography of Genetic Variants Browser (http://www.popgen.uchicago.edu/ggv).
(PDF)

**S7 Fig. Pairwise linkage disequilibrium ($r^2$) in A) Vietnamese, B) French and C) South African study cohorts (left panels) between 60 variants in a 30 kb window around rs17155120 (arrow) as compared to 1000G populations (right panels).**
(PDF)

**S8 Fig. Quantile-quantile plot of GWAS of resistance to *M. tuberculosis* infection in the 3 cohorts from Vietnam, France and South Africa (333 uninfected vs 616 infected subjects) ($\lambda$ = 1.06).**
(PDF)

**S9 Fig. Flowchart of the study.**
(PDF)

**S10 Fig. Principal component analysis of the Vietnamese study cohort.** Plot of the first and second principal components of the 720 individuals from the Vietnamese cohort after projection A) on the 1000 Genomes phase 3 population, B) on the East Asian 1000 Genomes Phase 3 populations only.
(PDF)

**S11 Fig. Principal component analysis of the French cohort.** Plot of the first and second principal components of A) 573 individuals from the French cohort and B) the 157 subjects subsequently analyzed in the GWAS, after projection on the 1000 Genomes phase 3 populations.
(PDF)

**S12 Fig. Principal component analysis of the South African cohort.** Plot of the first and second principal components of 374 individuals from the South African cohort after projection on the 1000 Genomes phase 3 populations.
(PDF)

**S13 Fig. Stacked histogram of the tuberculin skin test (TST) distribution, stratified by our infection definition among 1108 household contacts in Vietnam.** The dashed line represents a 5 mm cut-off for TST. Uninfected subjects (in yellow) were negative for TST and QuantiFERON-TB Gold In-Tube (QFT-GIT) and infected subjects (in blue) positive for both.
(PDF)

**S14 Fig. Definition of the GWAS phenotype in the household contacts (HHCs) study in France.** A) Distribution of the tuberculin skin test (TST) induration among 516 HHCs (mean induration when 2 screening visits were done). The dashed line represents a 5 mm cut-off. TST was considered negative when the skin induration was i) < 5 mm at both V1 and V2, ii) < 5 mm at V1, when only one visit was done. TST was considered positive when the skin induration was i) $\geq$ 5 mm at both V1 and V2, ii) < 5 mm at V1 and $\geq$ 10 mm at V2. B) Construction of a ROC curve based on the defined TST status to determine the optimal interferon-$\gamma$ release assay (IGRA) cut-off (175 pg/mL). C) Stacked histogram of the TST distribution stratified by our infection definition. Uninfected subjects (in yellow) presented a negative TST and a null production of IFN-$\gamma$. Infected subjects (in blue) presented a positive TST and a positive IGRA (IFN-$\gamma$ production > 175 pg/mL).
(PDF)

**S15 Fig. Definition of the GWAS phenotype in the family-based study in South Africa.** A) Distribution of the tuberculin skin test (TST) induration among 415 participants. The dashed line represents a 5 mm cut-off. B) Construction of a ROC curve based on the TST status to determine the optimal interferon-$\gamma$ release assay (IGRA) cut-off (20.9 pg/mL). C) Stacked histogram of the TST distribution stratified by our infection definition. Uninfected subjects (in yellow) presented a negative TST and a null production of IFN-$\gamma$. Infected subjects (in blue) presented a positive TST and a positive IGRA (IFN-$\gamma$ production > 20.9 pg/mL).
(PDF)

**S1 Table. Info score of the associated variants on 10q26.2.**
(PDF)

**S2 Table. Variants in the locus on chromosome 10q26.2 associated with resistance to *M. tuberculosis* infection in Vietnam (185 uninfected and 201 double positive infected subjects).**
(PDF)

**S3 Table. Frequency of rs17155120 minor allele T in 1000 Genomes phase 3 populations.**
(PDF)

**S4 Table. Variants associated with resistance to *M. tuberculosis* infection in the 3 cohorts from Vietnam, France and South Africa (333 uninfected vs 616 infected subjects) with P < 5.0 × 10−8.**
(PDF)

**S5 Table. GWAS cohort in Vietnam.**
(PDF)

**S6 Table. Risk factors associated with (i) double positive tuberculin skin test (5 mm cut-off) and QuantiFERON-TB Gold In-Tube results or (ii) double positive TST/QFT-GIT and pulmonary TB patients, compared to the reference group (double negative TST and QFT-GIT results) in household contacts in Vietnam.**
(PDF)

**S7 Table. Risk factors associated with both positive tuberculin skin test (5 mm cut-off) and positive interferon-γ release assay (IGRA) results compared to the reference group (negative TST and null IGRA) in household contacts in France.**
(PDF)

**S8 Table. GWAS cohort in France.**
(PDF)

**S9 Table. Risk factors associated with both positive tuberculin skin test (5 mm cut-off) and interferon-γ release assay (IGRA) results compared to the reference group (both negative TST and IGRA results) in the family-based study in South Africa.**
(PDF)

**S10 Table. GWAS cohort in South Africa.**
(PDF)

## Acknowledgments

We are grateful for the skilled technical assistance of Ho thi Ngoc, Binh Duong Center for Social Diseases Control, and Hanh La, Hospital for Dermato-Venereology, Ho Chi Minh City. We are indebted to the laboratory staff of the Hospital for Dermato-Venereology, Ho Chi Minh City, and the tuberculosis control teams of the Thu dau Mot, Thuận An, Ben Cat, and Di An districts. We sincerely thank all members of the community who participated in this work in South Africa and France and members of the lab of Human Genetics of Infectious Diseases for helpful discussions.

## Author Contributions

**Conceptualization:** Erwin Schurr.

**Data curation:** Jocelyn Quistrebert, Marianna Orlova.

**Formal analysis:** Jocelyn Quistrebert.

**Funding acquisition:** Jocelyn Quistrebert, Alexandre Alcaïs, Laurent Abel, Erwin Schurr.

**Investigation:** Le Thi Ton, Nguyễn Trong Luong, Nguyễn Thanh Danh.

**Methodology:** Jocelyn Quistrebert, Aurélie Cobat.

**Project administration:** Marianna Orlova, Christophe Delacourt, Eileen G. Hoal, Alexandre Alcaïs, Vu Hong Thai, Lai The Thành.

**Resources:** Gaspard Kerner, Le Thi Ton, Nguyễn Trong Luong, Nguyễn Thanh Danh, Quentin B. Vincent, Fabienne Jabot-Hanin, Yoann Seeleuthner, Jacinta Bustamante, Stéphanie Boisson-Dupuis, Nguyen Thu Huong, Nguyen Ngoc Ba, Jean-Laurent Casanova, Christophe Delacourt, Eileen G. Hoal, Alexandre Alcaïs, Vu Hong Thai, Lai The Thành.

**Software:** Gaspard Kerner, Fabienne Jabot-Hanin, Yoann Seeleuthner.

**Supervision:** Laurent Abel, Erwin Schurr, Aurélie Cobat.

**Validation:** Laurent Abel, Erwin Schurr, Aurélie Cobat.

**Visualization:** Jocelyn Quistrebert.

**Writing – original draft:** Jocelyn Quistrebert.

**Writing – review & editing:** Jocelyn Quistrebert, Marianna Orlova, Laurent Abel, Erwin Schurr, Aurélie Cobat.

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
