## [Decision Letter · Decision Letter 0]

5 Nov 2020

Dear Dr Cobat,

Thank you very much for submitting your Research Article entitled 'Genome-wide association study of resistance to Mycobacterium tuberculosis infection identifies a locus at 10q26.2 in three distinct populations' to PLOS Genetics. Your manuscript was fully evaluated at the editorial level and by independent peer reviewers. The reviewers appreciated the attention to an important topic but identified some aspects of the manuscript that should be improved.

We therefore ask you to modify the manuscript according to the review recommendations before we can consider your manuscript for acceptance. Your revisions should address the specific points made by each reviewer.

[LINK]

Yours sincerely,

Cathy Stein

Guest Editor

PLOS Genetics

Hua Tang

Section Editor: Natural Variation

PLOS Genetics

The reviewers generally had a very positive assessment of this manuscript, with mostly points of clarification requested. Please address the data access issue; it is my understanding that PLoS Genet will allow data sharing as you have proposed, but it must be clearly justified.

Reviewer's Responses to Questions

**Comments to the Authors:**

Reviewer #1: Thank you for your interesting study and well written manuscript. Overall,

I found the report clear and comprehensive. I have a few comments that I feel would improve the

manuscript, as follows:

Line 161-167: It is possible that LTBI has different genetic etiology compared to PTB and yet you group the two together as infected. You do state that you additionally ran the GWAS in the subset with PTB removed and you show that the chromosome 10 SNPs remain significant. It would be interesting to see the GWAS results for the PTB removed subset, at least to see a manhattan plot comparison.

Do the French and south African cohorts include PTB subjects combined with the TST+/IGRA+ group?

Previous Results: What are the results for variants or regions previously identified. In the introduction you reference the studies and findings, but do not show what the association results looked like in your current study. I believe it would be informative to see what the previously identified results were in your current study.

Line 399-403: Did you include PCs in your regression model? You check the association between many covariates and the phenotype, but the list of covariates does not include PCs.

Line 399-403: You check for covariates and their association with TST+/IGRA+ vs. TST-/IGRA null. Do you check the association between the covariates and the combined phenotype (TST+/IGRA+ or PTB vs. TST-/IGRA null)?

Reviewer #2: In this paper, Quistrebert et al performed a GWAS of resistance to TB infection, using 3 distinct cohorts. First, they evaluted a sample of Vietnamese household contacts from individuals from Vietnam, followed by validation in two separate cohorts, France and South Africa. In each cohort, genetic results were adjusted for ancestral genetic architecture, age, and sex. They found a locus on 10q26.2 as strongly associated with resistance to either QFT or TST in the Vietnamese population, and validation at lower significance in the other populations. These data represent an interesting addition to the growing field of TB genetic studies, with the added novelty of evaluation of the unique "TB resister" phenotype. Therefore these results are of a good deal of interest to the field. I have some minor issues to address before publication, but generally, this is a quite well written and methodologically sound manuscript.

Major issues;

Although there was description of how resistance was defined, it would be nice to have a bit more clarity on this score, as the longer the duration of exposure to TB, the more powerful the genetic association of the phenotype becomes. I wonder if there was there a determination of an exposure score in the cohorts, as done in the Stein Ugandan cohort? Is there a way to define the amount of time under exposure to Mtb more than estimated coughing time?

Minor issues: It appears that many of the individuals in Vietnam cohort developed active TB as compared to other HHC. How does this number compare to overall measures of progression from those individualas exposed to TB? If it is higher than expected, does that mean anything for the future analysis?

Reviewer #3: The manuscript by Quistrebert reports an association of a region on chromosome 10q26 with resistance to tuberculosis in highly-exposed individuals, replicated in three cohorts of diverse racial/ethnic backgrounds. The study is well-designed, with rigorous analysis and quality control checks. Annotation of the association by public expression database results suggests that the variants associated with resistance might modulate a Th17 immune response.

In general, the paper is well written and the results and interpretation are convincing. There are a few points that the authors should address to improve the impact of the paper:

1) The definition of “highly exposed” seems to vary a bit across the three cohorts, with two appearing to depend largely on co-residence and long term exposure to a PTB case (Vietnam & France) but one not requiring such exposure (S. Africa). Could the effort to quantify or establish exposure be better described and any effort to harmonize across studies be outlined? This seems less necessary with the consistency of results, but the results in SA are the weakest, and this might be due to misclassification of individuals as “resistant, highly exposed” and attenuation of the association accordingly.

2) There is no discussion of the role of HIV in these three cohorts, other to note in the introduction that seroprevalence is low overall in Vietnam. Was HIV status known in these individuals? If not, this should be discussed.

3) The use of an uncorrected p-value threshold of 0.05 for replication should be discussed and justified, since multiple variants were tested. While LD was very strong in the Vietnam sample (thus suggesting this is really one test) it is weaker between some marker pairs in the other two samples, suggesting some correction might be necessary. To be sure, Bonferroni correction is too stringent, but an estimate of the number of independent tests is possible, and perhaps advisable here.

4) While the authors indicate that data are publicly available, the doi provided lists summary statistics for the 3 datasets. PLoS Genetics data availability policy states that large-scale data (such as GWAS) should be made available in a public repository upon publication. The authors should discuss whether this is possible, and if not, explain why only summary statistics will be provided.

**Have all data underlying the figures and results presented in the manuscript been provided?**

Reviewer #1: Yes

Reviewer #2: Yes

Reviewer #3: Yes

PLOS authors have the option to publish the peer review history of their article (what does this mean?). If published, this will include your full peer review and any attached files.

Reviewer #1: No

Reviewer #2: No

Reviewer #3: No

---

## [Decision Letter · Decision Letter 1]

2 Feb 2021

Dear Dr Cobat,

We are pleased to inform you that your manuscript entitled "Genome-wide association study of resistance to Mycobacterium tuberculosis infection identifies a locus at 10q26.2 in three distinct populations" has been editorially accepted for publication in PLOS Genetics. Congratulations!

Yours sincerely,

Cathy Stein

Guest Editor

PLOS Genetics

Hua Tang

Section Editor: Natural Variation

PLOS Genetics

Comments from the reviewers (if applicable):

Reviewer's Responses to Questions

**Comments to the Authors:**

Reviewer #1: Thank you for your response to my comments regarding the manuscript. Your responses, including additions to the manuscript were sufficient in addressing the questions I posed and I believe beneficial to the manuscript as a whole. Your responses that were to clarify questions I posed, but did not require additions to the manuscript were appreciated as well.

Reviewer #2: Thank you for the thorough response. I believe this paper will provide a great addition to the understanding of host genetics of tuberculosis

Reviewer #3: The authors have addressed the points raised on the previous review, and I have no further suggestions for changes to the manuscript.

**Have all data underlying the figures and results presented in the manuscript been provided?**

Reviewer #1: Yes

Reviewer #2: Yes

Reviewer #3: **No: **While summary statistics are provided, the underlying raw data may not be placed in a public repository due to restrictions in the consent obtained from participants. Data may be requested from the authors and sharing can take place through a collaborative effort. Whether this is sufficient compliance with the journal data availability policy, or whether an exception should be made in this case, is a decision for the editors to make.

PLOS authors have the option to publish the peer review history of their article (what does this mean?). If published, this will include your full peer review and any attached files.

Reviewer #1: No

Reviewer #2: **Yes: **Javeed A. Shah

Reviewer #3: No

**Data Deposition**

http://datadryad.org/submit?journalID=pgenetics&manu=PGENETICS-D-20-01562R1

**Press Queries**

---

## [Editor Report · Acceptance letter]

28 Feb 2021

PGENETICS-D-20-01562R1 

Genome-wide association study of resistance to *Mycobacterium tuberculosis* infection identifies a locus at 10q26.2 in three distinct populations 

Dear Dr Cobat, 

We are pleased to inform you that your manuscript entitled "Genome-wide association study of resistance to *Mycobacterium tuberculosis* infection identifies a locus at 10q26.2 in three distinct populations" has been formally accepted for publication in PLOS Genetics! Your manuscript is now with our production department and you will be notified of the publication date in due course.

With kind regards,

Alice Ellingham

PLOS Genetics

On behalf of:
